# Piezoelectric Micromachined Ultrasonic Transducers with a Cost-Effective Bottom-Up Fabrication Scheme for Millimeter-Scale Range Finding

**DOI:** 10.3390/s19214696

**Published:** 2019-10-29

**Authors:** Guo-Hua Feng, Hua-Jin Liu

**Affiliations:** Department of Mechanical Engineering, National Chung Cheng University, Chiayi 621, Taiwan; blueforest1029@gmail.com

**Keywords:** piezoelectric, ultrasonic transducer, resonance, micromachining, rangefinder

## Abstract

This study proposes a novel piezoelectric micromachined ultrasonic transducer (PMUT), fabricated on a metal foil. Using a bottom-up, cost-effective micromachining technique, the PMUTs made of electrodes, a piezoelectric film, or electrode-sandwiched structures with versatile patterns were implemented on a large-area foil thinner rather than regular paper. The proposed microfabrication facilitated the PMUT to be able to generate ultrasonic waves with fundamental and harmonic resonances. The fourth-order resonances of the fabricated PMUT functionally operated at an ultrasonic spectrum of approximately 30 kHz as an ultrasonic emitter. The developed PMUT was paired with a microelectromechanical system (MEMS) microphone module for range-finding applications in the range of several tens of millimeters. A signal-processing scheme was developed to extract the representative pattern from the acquired signals that were emitted and received. The pattern enabled finding the distance between the PMUT and the microphone using time-of-flight and strength-variation technology. The developed PMUT-microphone pair demonstrated its range-finding performance, displaying an error of less than 0.7% using the time-of-flight method.

## 1. Introduction

Currently, ultrasound instruments operate at frequencies ranging from tens of kilohertz to thousands of megahertz. They have become effective and important tools in many applications, such as medical imaging, therapeutic ultrasound, non-destructive evaluation, underwater sonar, particle and cell manipulation, and ultrasonic actuation [1,2,3,4]. In addition, an ultrasonic rangefinder has been broadly harnessed for the gesture interface of handheld gadgets, vehicle-driving systems to avoid collision, and robots for detecting obstacles during operations [5,6,7,8,9].

Micromachined ultrasound transducers (MUTs) display great potential in dealing with the aforementioned issues. As the fabrication process is relatively uniform, it is possible to integrate arrays with a high element density, small element size, MUT arrays, and external circuits in a chip for packaging. Currently, capacitive micromachined ultrasonic transducers (CMUTs) and piezoelectric micromachined ultrasonic transducers are being studied by an increasing number of research groups [10,11,12,13]. The deflection of the flexural mode of the piezoelectric micromachined ultrasonic transducer (PMUT) mainly depends on the lateral strain of its piezoelectric film while an AC voltage is applied [14,15,16].

The piezoelectric film and structure of the PMUT are two dominant factors associated with the sensitivity and performance of PMUTs. Although the improved bandwidth of PMUTs and coupling with air have been progressively reported [17,18,19], the construction of PMUTs mainly utilizes a top-down fabrication process. For instance, several layers are deposited to form active PMUT structures on a silicon wafer. Following this, a cavity is created beneath the structure of the PMUT, either using backside etching or release from a silicon-on-insulator (SOI) wafer, to construct PMUTs. The residual stress control of each deposited layer is critical in order to obtain optimized PMUTs from this process [20].

Existing commercial range-finding sensors invariably function by emitting laser, infrared light-emitting diode (LED), or ultrasonic wave signals to the target. The distance from the target is realized by receiving the returned signal and reading the way in which it has changed on its return. The intensity change of the returned signal or the time taken by the signal to return is evaluated to find the corresponding distance. Among the versatile distance-sensing technologies applied in the Internet of Things (IoT), Industry-4.0-related markets, autonomous vehicles, etc., the most affordable resources include LEDs, light detection and ranging (LIDAR), ultrasonic, and vertical-cavity surface-emitting laser (VCSEL) modules [21,22,23,24]. The price of these products ranges from several tens to several hundred US dollars, depending on their specifications.

The commercial ultrasonic distance sensor is typically based on the core vibration element, which is a piezoelectric ceramic material, such as lead zirconate titanate (PZT) [25,26]. The piezoelectric element operates at resonance through the structure vibration mode, which depends on the configuration of the entire ultrasonic emitter or sensor design. Existing ultrasonic distance sensors operate with a fixed operation frequency depending on the application [27,28].

This study presents a bottom-up PMUT fabrication technique. It starts with a PZT film with a near-zero residual stress quality grown on a 5-µm thick metal foil and utilizes stereographic 3D printing to realize the designed base of the PMUT. The merit of the proposed PMUT consumes less current and is cost-beneficial. The novel 3D printing fabrication has significant advantages, such as being flexible and custom-made. The pair formed with PMUTs and a MEMS microphone exhibits an even smaller distance detection capability than the recent VCSEL-technology-based distance sensor (less than 10 mm), and a comparable resolution (less than 1 mm). More details are described in the following sections.

## 2. PMUT Design and Finite Element Analysis (FEA) Simulation

### 2.1. Structural Design of the PMUT

There were two main design concepts involved in the construction of the proposed PMUT. One was the foil containing the piezoelectric element and the other was the PMUT base, which was utilized to determine the vibrational area of the PMUT. In addition, the design consideration had two aspects: (1) The active foil was designed as a large circular geometry with four beams connected to the anchor region. Compared to a common four-edge anchored square membrane, this design allowed for the active membrane to possess a larger displacement through its less-fixed boundary condition. (2) Considering this prototype design was manually assembled onto the 3D printed base and setting the center of circular foil was done in alignment with the opening center of 3D printed base, we thus designed the entire structural foil to be approximately 2 mm × 2 mm for relatively easy implementation in this study.

The study started with a geometrical delineation on a titanium foil of 5 µm as the substrate of the PMUT. Since the proposed fabrication process could have a versatile substrate design, we chose a circular plate connected to four symmetrical beams and fixed the boundary conditions for the outer edges of the four beams. The designed geometrical parameters of the substrate was determined using the FEA analysis, as mentioned in the next section. A circular shape with a diameter of 1.5 mm was drawn in the XY plane, with its center located at the origin of this plane (Figure 1). Four lines offsetting the *X* and *Y* axes were plotted to define the beams. The offset was set at ±0.3 mm away from the *X* and *Y* axes. Hence, eight points intersected with the edge of the circular plate, with each line contributing two points. The corner points of the beams were defined as points on the offset lines at a distance of 0.4 mm away from these intersection points. Consequently, the corner points, intersection points, and circular arcs determined the contour of the substrate.

To construct the PMUT on the substrate, two functional layers were deposited on the substrate. A hydrothermal PZT layer of 4 µm was formed on both sides of the substrate. The top silver electrode layer of 1 µm was deposited on top of the PZT layer with a shadow mask to extract the electrode pattern 100 µm away from the edge of the substrate.

The PMUT base was responsible for securing the foil containing the designed piezoelectric element with a specifically selected area. The base design was versatile and implemented using stereography. The portion of the foil attached to the base created a joined rigid body. The non-attached part of the foil resulted in a freestanding structure surrounded by the opening of the base. Because the PMUT base had a greater thickness than the foil containing the piezoelectric element, the freestanding portion of the foil could be considered to have a fixed boundary condition, bounded by the edge of the base opening. The PMUT base designed in this study was a cuboid with a length of 10 mm. The base opening was designed as a square, located at the center of the cuboid, with an opening length of 2.17 mm, which was the distance between the two farthest edges of the PMUT designed on the foil.

### 2.2. ANSYS Simulation

ANSYS was used to perform modal analysis and harmonic response analysis on the designed PMUT. The structural model was set along the edges of the four beams with a fixed support boundary condition. Since each layer of the PMUT was much thinner than its lateral dimension, a single layer with an equivalent Young’s modulus, density, and combined thickness was used to find the resonances. The equivalent Young’s modulus and density using the parameters in Table 1 was calculated in accordance with Feng and Chen [29]. We have used a composite structure with multi-layer parameters for the ANSYS model, but the large aspect ratio of the structure, such as the 1-µm-thick electrode layer with the length of more than 2 mm made it difficult to perform the simulation difficult with a limited mesh number performed in a personal computer. Using the equivalent parameters of the composite structure, we could obtain a good match between the model and experimental results, as described below.

The first four resonant frequencies were 11.211, 22.627, 22.629, and 31.576 kHz. The corresponding mode shapes are displayed in Figure 2a–d. The maximum displacement of the first resonance occurred at the center of the structure. The second and third mode shapes exhibited a phenomenon of maximum displacement, which occurred at two different edges of the opposite quarter-circle of the designed PMUT. Ideally, due to the symmetry, the frequency of the second and third mode should be identical, hence it is possible to view the minor difference as a numerical error. However, in a practical test or operation, several situations might lead to less than ideal results. For instance, the fabricated PMUT membrane might not be perfectly symmetric, causing a frequency splitting of the two resonances [30]. The amplitudes of each resonance also depend on the effective location of the excitation. The fourth mode shape occurred at a single frequency. The ensuing mode shape was supposed to be relatively stable during the operation.

In the following study, the first mode (approximately 11 kHz) was investigated for the range-finding experiment. The receiving signal was easily disturbed by the surrounding noise due to this operating frequency being within the audible range. Therefore, we chose the fourth mode, which operated above 20 kHz, as the operating frequency for our range-finding study. Additionally, since rare papers have reported the performance and likely application of PMUTs operated at the higher resonance modes, we tried to demonstrate the developed PMUT could operate at the fourth mode resonance as a rangefinder in this study.

## 3. Device Fabrication

### 3.1. Realization of the Foil Substrate with Designed Patterns

A 5-µm-thick titanium foil was chosen as a substrate. The PMUT array was fabricated on this foil with proper dimensions. A photomask of nine elements, with each following the aforementioned design guideline, was created (Figure 3). The square element had a side length of 2.7 mm. Each element had a circle in the middle with a radius of 0.75 mm. Four beams with a width of 0.8 mm were connected to each circle. A patterned foil substrate was fabricated using a micromachining process. The foil was secured on a glass plate with four pieces of tape glued to its corners. A 2-µm-thick photoresist AZ5214 was spin-coated onto the foil. After exposure to UV light and development, the designed geometry of the photomask was transferred to the photoresist layer.

The wet etching process was administered at room temperature. A diluted multi-metal etching solution was used to create openings on the foil. An arc-shaped curve around the corner regions of the element was observed because of the wet etching. The lateral etching effect caused the opening to be slightly larger than the designed photomask. The patterned titanium foil was removed from the glass plate and rinsed with deionized (DI) water, to be served as the PMUT substrate for subsequent piezoelectric film deposition.

### 3.2. Hydrothermal Growth of the PZT Film

To perform hydrothermal synthesis of the PZT film, an aqueous solution of titanium dioxide was first prepared. A 2.87 g mass of TiO_2_ powder was added to 350 g of DI water, and mixed by stirring for 10 min. Subsequently, 27.07 g of zirconium oxychloride and 50.07 g of lead nitrate were added to the processed solution and stirred for 1 h. Then, 200 mL of 10 M potassium hydroxide solution was arranged and mixed with the solution containing TiO_2_/Zr. This highly alkaline solution effectively aided the growth of the PZT film on the titanium foil. The molar ratio of Pb/Zr/Ti in the solution was 4.3:2.3:1. The color of the solution became reddish brown after stirring for 1 h.

The hydrothermally grown piezoelectric film was executed as follows [31]. The patterned titanium foil was anchored on a metal holder and the prepared precursor solution was poured into a homemade steel autoclave.

The autoclave was composed of a heating module, pressure gauge, temperature probe, pressure relief valve, a top cover with a rotation mechanism, and a low-speed motor. The heating module was surrounded by the lower chamber of the autoclave. As the temperature probe detected the temperature of the processed solution and fed the signal back to the proportional, integral, derivative (PID) controller, the error of the set temperature could be confined to be within 2 °C. The holder with the titanium substrate was placed inside the autoclave, such that the processed titanium foil was fully immersed in the solution when the autoclave was hermetically closed.

Different PZT film properties can be obtained by varying the temperature setting during deposition. In this study, the prepared solutions were heated inside the autoclave from room temperature to the temperature setting of 180 °C at a rate of 5 °C/min. The processing temperature was controlled at 180 °C when it reached a steady state and was maintained at a constant pressure to obtain a quality PZT film. The process temperature was below the Curie point of a PZT film (usually above 200 °C). The designated process temperatures were maintained for 40 h. Subsequently, the heating function was turned off to allow the solution to cool to room temperature, and the titanium substrate with the developed PZT film was removed from the autoclave.

Due to the simultaneous growth of the PZT film on both sides of the titanium foil and the well-controlled processing condition, the large-sized foil coated with a 10-µm-thick PZT film exhibited almost zero residue stress.

Figure 3 displays the fabricated PMUT film element array on the processed titanium foil. Each PMUT element was constructed with the patterned foil substrate, PZT film, and silver electrode. This thinner-than-hair structure could be attached to curved surfaces without cracking, which demonstrated the fabrication process of the superior PMUTs with a near-zero residual stress.

The major part of the fabrication process was concerned with a titanium foil with a free boundary condition, which allowed the residual stress to be easily released through its geometrical change. The photolithography process to delineate the pattern on the foil and the following wet etching to produce desired openings on the foil, as well as the hydrothermal grown PZT on the foil and sputtering electrode, were all processed under the condition of the foil remaining flat. If a significant residual stress was generated in these processes, the processed foil was easily curved or the cracks were easily found in the deposited film on the foil. However, we did not observe these problems during the processes. As for the assembly process that involved gluing the membrane device to the 3D printed base, the likely residual stress was produced at the interface between the inactive part of the foil and anchored base, i.e., not associated with the performance of the PMUT. The level to which residual stress affected the performance of the PMUT could be an issue for further investigation. The goal of our study was to minimize the residual stress effect within the active film of the PMUT such that it benefited the PMUT performance.

To make the designed structural dimensions within an acceptable range using the proposed bottom-up method, this strongly depended on the equipment/aligner accuracy, the process parameter control, and how proficient the operator handled the photolithography, wet etching, and electrode sputtering processes. For a well-trained engineer with precision equipment to fabricate the foil containing piezoelectric elements for the PMUT, achieving a 1–3% error should not be difficult. In this study, the fabricated error of the structural dimension of the PMUT membrane was approximately 5%.

### 3.3. From the PZT Foil Element to the PMUT Device

A sandwich structure of electrode/PZT/electrode was constructed to make the PZT foil element functional. The titanium foil, which was coated with the PZT film, was utilized as the bottom electrode, and the deposited metal layer on the PZT film was used as the top electrode. A 10-µm-thick shadow mask fabricated with the aforementioned micromachining technique was created. Completely cross-shaped patterns were created in an array form that covered the PZT-coated foil with a proper alignment. Silver was chosen as the material for the top electrode. A 1-µm-thick silver film was sputtered to define the active region. The bottom electrode was revealed by selectively etching the unwanted PZT film on the designed region of the PZT-coated foil.

Next, the adhesive bonding method was used to connect the processed foil containing the PZT elements to a 3D printed base. The base geometry followed the aforementioned design and was plotted using Solidwork software for production. The 3D printing machine was manufactured by XYZ printing Co. with model no. Nobel 1.0 A (New Taipei City, Taiwan). It had an *XY* axis resolution of 130 µm and layer thickness setting of 25, 50, or 100 µm. A thin epoxy layer was applied as an adhesive to the top surface of the finished cuboid with the square opening. The backside of the processed foil, which was the side without the silver electrode, was then carefully aligned and attached to the base such that the PMUT was complete and ready to be tested (Figure 4). The leads were secured on the prototype of the PMUT with two kinds of adhesive. The conductive silver paste was first applied to connect the lead with the titanium foil for electrical signal delivery. Then, the non-conductive epoxy was covered with the silver paste region. This allowed for the lead to be firmly secured on the PMUT to facilitate the subsequent testing. This scheme allowed the piezoelectric foil that was connected to the base to maintain the property of released residual stress, which was an advantage compared to conventional PMUT fabrication.

## 4. Experimental Setup

### 4.1. Characterization of the Piezoelectric Foil Response

First, an experiment was performed to find the resonances of the fabricated PMUT. The input was set as a sinusoidal wave, with the frequency ranging from 10–30 kHz and an amplitude of 20 V peak-to-peak (sweep: 1 s; return: 0 s; interval: 1 ms setting on the function generator Textronix AFG3022, Beaverton, OR, USA). A laser displacement meter (Keyence Co., Osaka, Japan) with a sampling rate of 392 kHz was employed to obtain the displacement of the PMUT. A laser spot was emitted onto the center of the circular foil of the PMUT (Figure 5a). The signal of each input setting was acquired for 1 s using the NI 6351 DAQ card. Subsequently, the recorded data was processed with MATLAB (TaraSoft Corp., Taipei, Taiwan) to analyze its frequency response.

The second experiment investigated the displacement of the fabricated PMUT, which was operated at the resonant frequency of interest. The resonant frequency of operation was obtained from the above analysis. Different amplitudes of actuation voltages were used to drive the PMUT, and the same laser displacement meter setup (as mentioned above) was applied to receive the signals. The acquired signals were processed with a bandpass filter in order to obtain the displacement data.

For IoT applications, the power consumption could be an importance issue. Although the input amplitude of 20V peak-to-peak was delivered to characterize the developed PMUT, the small consuming current would be an advantage for this piezoelectric actuator. The power consumption of the PMUT could be another topic for subsequent study.

### 4.2. Characterization of the Fabricated PMUT Pairing with the MEMS Microphone as a Rangefinder

A commercially available MEMS microphone IC chip was used as an ultrasound wave receiver to perform the experiment of pairing the fabricated PMUT and MEMS microphone as a rangefinder. The selected MEMS microphone was InvenSense ICS-40181, which was an analog device with a high signal to noise ratio (SNR) and enhanced (radio-frequency) RF immunity. The printed circuit board (PCB) board was implemented to integrate this microphone and an amplifier AD8541 as a module, to receive the signals generated by the PMUT. According to the datasheet ICS-40181 provided by the manufacturer, the frequency response from 10 Hz to 30 kHz has been measured. In the range-finding experiment, the developed PMUT operated at 29.85 kHz to emit the ultrasonic wave and was paired with the ICS-40181 microphone to receive the ultrasonic wave. We found that the dynamic range of the selected microphone could respond beyond 30 kHz based on the analysis of the acquired signals.

The range-finding experiment was executed as follows (Figure 5b). An L-shaped 3D printed holder was anchored on the sidewall of a XZ precision stage. The stage was produced by Newport Corp. (Irvine, CA, USA); its motion could be controlled using a computer with a piezoelectric actuation mechanism. Its travel resolution was 20 nm and the maximum stroke was 3 cm. The fabricated PMUT was anchored at the bottom of the L-shaped structure. A spacer was attached to the base of the PMUT such that the initial distance between the piezoelectric foil of the PMUT and the sound hole of the MEMS microphone was set at 5 mm.

The homemade PCB board with the microphone chip and associated electrical components was mounted on a table for testing. To prevent ultrasonic waves produced by the PMUT from scattering into the sound hole and causing unwanted signals from being received, sound absorbent cotton (3M Thinsulate Acoustic Insulation SM600, St. Paul, MN, USA) was utilized to cover the entire area, except the sound hole. The sound hole was then aligned with the piezoelectric foil center of the PMUT for measurement.

The fourth resonance of the fabricated PMUT was chosen as the operation frequency during testing. This was because it was relatively easy for the fundamental resonance of approximately 11 kHz to interfere with the noise existing in the laboratory that was caused by various instruments and equipment. In addition, as the strength of the frequency spectrum for the second and third resonance was less than that of the fourth resonance in the characterization of the piezoelectric foil vibration, the range-finding experiment was conducted utilizing the fourth resonance as the operation frequency.

The experiment started with a specified sinusoidal electrical signal emitted from the digital to analog (D/A) port of NI-6351 DAQ card, through the Labview (National Instruments, Taipei, Taiwan) setting. An identical DAQ card with a sampling rate of 1 MHz acquired the received analog signal from the microphone module. The parameter setting on the Labview to configure the simulated signal for driving the PMUT is specified below. Signal type: sine; frequency: 29,411 Hz; amplitude: 1.5 V; phase and offset: 0. Timing frame: samples per second: 106 Hz; number of samples: 34. The signal was then amplified by 20 times with a piezo linear amplifier (Model No. EPA-104, Piezo.com, Woburn, MA, USA). This voltage selection was considered the maximum sustainable value that the PMUT could be operated under.

## 5. Results and Discussion

### 5.1. Frequency and Displacement Response of the Fabricated PMUT

Figure 6 illustrates the frequency response of the PMUT within the frequency range of 10 to 30 kHz, with a sweeping span of 20 kHz. This frequency spectrum was derived by processing the raw data acquired from the laser displacement meter. Using a fast Fourier transform (FFT) with a 392 k data number and sampling frequency of 392 kHz, a resolution of 1 Hz was attained. A dominant peak was exhibited at about 11 kHz, which was close to the simulation results of 11.211 kHz. A relatively small peak could be observed at approximately 22–23 kHz, which could be considered the second and third resonances. Another obvious peak possessing the strength of about one-tenth of the first resonant peak was observed at around 29 kHz, which could be viewed as the fourth resonance. It corresponded to the simulation result of 31.576 kHz. If the PMUT was driven further with a smaller sweeping frequency range, more accurate resonant frequencies were obtained. For example, when the sweeping range was confined between 11 and 11.6 kHz, the first resonance occurred at 11.45 kHz. The fourth resonance displayed the maximum value at 30.05 kHz while the sweeping frequency was specified in the range of 28.5–30.5 kHz.

### 5.2. Displacement of Piezoelectric Foil at Resonance for a Varied Input Voltage

The resonances were found experimentally through the frequency response of the fabricated PMUT. A significant property of the PMUT was the maximum displacement normal to the surface at its operation resonance. Figure 7a shows the displacement of the PMUT operated at 11.45 kHz with a driving sinusoid wave of 20 V peak-to-peak. This displacement result was obtained with the acquired signal from the laser displacement meter after processing with a second-order Butterworth bandpass filter. The center frequency was set at 11.45 kHz and the cutoff frequencies were 1.045 and 21.45 kHz. The maximum displacement achieved was approximately 8.6 µm. Using a similar signal-processing method, the displacements of the PMUT were measured with different input actuation voltages. Figure 7b exhibits the linear trend of maximum displacements from 10 to 20 Vpp. For an increment of 1 Vpp, the displacement was about 0.24 µm.

The displacement of PMUT, operated around the fourth resonance, was also characterized. The frequency of 29.85 kHz was chosen instead of 30.05 kHz, at which the maximum peak in the frequency spectrum existed because of the two reasons mentioned below. In terms of practical intervals in operating frequencies, this frequency maintained a maximum increment up to 200 Hz. Instead, a less than 20 Hz decline of 30.05 kHz would lead to the resonance strength getting lower than 29.85 kHz. This meant that the PMUT that operated at 29.85 kHz would be more stable. Since the fourth resonance mode would be used for the ranging-finding experiment, PMUT operation at 28.95 kHz was a better choice. We selected the 29.85 kHz instead of 11 kHz as the operating frequency in the range-finding experiment due to the operation of 11 kHz in audible sound range being easily interfered with by the surrounding noise. For later work, it could be worthwhile to re-design the first resonance mode to be within the ultrasonic range to obtain a better signal to noise ratio. In the current study, one major contribution was the demonstration that the higher harmonic resonance was still functional for range-finding applications.

Similar signal processing as the fundamental frequency experiment was executed at 29.85 kHz, instead of the filter’s center frequency setting, and cutoff frequencies at 26.85 and 32.85 kHz. Figure 7c shows the resulting signal riding on a small fluctuation. To evaluate the amplitude of displacement, the envelope of the filtered signal was utilized to find the amplitude value. The upper- and lower-line points of the envelope, which were derived from the 392k sampling points, were averaged respectively, and the difference of these two averaged values was calculated as the result of the displacement amplitude. Furthermore, the PMUT driven with a varied voltage ranging from 10 to 20 Vpp was characterized. The resulting amplitudes also exhibited a near-linear relation, but it was smaller than that operated at the fundamental frequency (Figure 7d). The increase in amplitude per volt was approximately 0.08 µm/V.

### 5.3. The Raw Data of the Received Signals in the Microphone Module

The designed single waveform of the input signal to actuate the PMUT is shown in Figure 8a. A 29,411 Hz sinusoidal wave was created using 34 points and a sampling rate of 1 MHz was used. Two DC signals, each of 0 V and duration of 0.01 s (104 points at a sampling rate of 1 MHz), were added to the front end and backend points of the sine wave. Although the result of the characterization of the developed PMUT was 29.85 kHz for the fourth resonance operation, the practical operating frequency of a digital signal given by the experimental setup for delineating a sine wave was limited. This was because of the synchronous setting of the DAQ card for analog-to-digital and digital-to-analog conversions with an identical sampling rate of 1 MHz. Therefore, if 33 or 35 points were used with an interval of 10^−6^ s between any two neighboring points to define the sine wave, the resulting frequency would be 30.303 or 28.571 kHz, respectively. In comparison, the frequency value deviating from 29.85, 30.303, and 28.571 kHz were larger than 29.411 kHz. Hence, 34 points were set to generate the digital sine wave for testing.

While the PMUT emitted ultrasonic waves, electrical excitation was provided by repeatedly sending the designed single waveform as a continuous signal. Figure 8b shows the acquired signal from the microphone module without any data processing, at a distance of 10 mm away from the PMUT. The received ultrasonic waveform from the microphone module could be easily monitored behind the sine wave excitation.

### 5.4. Extracting the Representative Data Pattern from the Received Signal

The characteristics of the received signal from the microphone module, caused by a single ultrasonic wave emission, exhibited an interesting pattern. Several waves with periods similar to the emitted wave appeared and were then followed by noise (random periodic signals similar to the signal portion before the sine wave excitation (Figure 8b). This is discussed in detail below.

The time interval of the signal portion with a similar period was calculated to be approximately 0.0008 s. Therefore, a period of 0.020034 s was long enough for the excited sine signal to avoid the received signal, due to the excitation that followed, interfering with the current received signal. Actually, the period of the excitation signal could further be reduced in future studies. For instance, if the period of the excitation signal was shortened to 1 ms, the response time for measuring the distance could be improved by 50 times. The received signal exhibiting multiple waves or a ringing phenomenon could be explained by the resonant effect of the membrane in the MEMS microphone. Based on the datasheet of ICS40181, the manufacturer measured the typical frequency response from 10 Hz to 30 kHz, which displayed a resonance peak at approximately 23 kHz and a −6 dB bandwidth of ~14 kHz. The received wave of 29.85 kHz was operating within its resonance band. Hence, this caused the membrane of the microphone to oscillate and operate at a frequency that was not just the same as the input ultrasonic wave frequency, but coincided with the peak frequency of the microphone membrane resonance. This could be elucidated based on the data analysis below.

The raw signal was filtered with a third-order Butterworth bandpass filter with the cut-off frequencies set at 19,411 and 31,411 Hz. After processing, the data exhibited a smoother curve and the bias voltage attained a value of zero. This allowed us to easily extract a significant pattern from the signal (Figure 9a).

It was significant that if the processed signal in Figure 9a was taken at 66 successive points, ranging from the starting moment of the input signal and examined at its dominant frequency, it was found that the frequency had some peaks, in which 24.24, 31.82, and 46.97 kHz were approximately three to five times beyond the noise level. However, if 128 points of processed signal were chosen from the beginning of the input signal, strong peaks with much better signal-to-noise ratio were observed at 24.03, 31.78, and 39.53 kHz (Figure 9b,c). The two peak frequencies of approximately 24.03 and 31.78 kHz were significant in this study because the former was related to the fundamental resonance of the microphone membrane and the latter could be considered to be caused by the PMUT input signal. The appearance of the fundamental resonance could also be attributed to the excitation frequency of the input signal, which was still within the resonance band.

According to the analysis in Figure 9, we verified that the receiving signal from the time interval between the 66th and 128th points contained the strong input signal frequency instead of a noise signal. The first largest peak of this receiving signal after the input signal could also be found within this interval. We then used the common scheme to determine the time of flight by calculating the difference between the start time of the input signal and the first largest peak time of the receiving signal.

Figure 10 presents the results of the measured data for different distances after the bandpass filter processing. We tried to find the largest peak time of the wave to perform the analysis. Since the data points might not be located at the exact peak position of the wave, a curve-fitting method was utilized to find the peak time (Figure 11). The points of the first wave with positive amplitudes was employed to fit a fourth-order polynomial curve. A maximum y-coordinate (*y_a_*) and a corresponding x-coordinate (*x_a_*) of the curve could be obtained by differentiating the fourth-order polynomial and setting it to zero. *x_a_* was related to the travel time from the PMUT to the microphone, which allowed for evaluation of the distance between the PMUT and the microphone. *y_a_* represents the amplitude of the acquired signal at a fixed strength of the input signal. *x_a_* and *y_a_* were the two features used for range finding.

### 5.5. Increasing the Accuracy of the Range-Finding Measurement

To verify the function of the developed PMUT and MEMS microphone module pair serving as the rangefinder, *x_0_* was defined as the data number of a certain input signal *S_i_* at the initial position. The signal *S_i_* then triggered the pattern on the microphone-acquired signal, from which, the features of *x_a_* and *y_a_* were extracted, as mentioned above. Since the input signal to drive the PMUT and the signal from the microphone were simultaneously acquired, the data difference $\Delta x_{a}=x_{a}-x_{0}$ of these two signals was calculated and converted to a time interval $t_{a}=\Delta x_{a}\cdot f_{s}$, where *f_s_* represents the sampling frequency. Therefore, the relation between *t_a_* and *y_a_* could be found as a function of the measured distance *d_m_* (between the PMUT and microphone). The relationship of *t_a_* versus *d_m_* was related to the time-of-flight principle, and *y_a_* versus *d_m_* was associated with the strength variation technology for range finding.

Moreover, in order to reduce the uncertainty of $\Delta x_{a}$, 10 data sets were successively picked starting from an arbitrarily selected *x_a_*. Their average value was calculated using:(1)$$\bar{\Delta x_{a}}=\overset{10}{\underset{i=1}{\sum }}\Delta x_{a,i}/10$$

Similarly, the average value of *y_a_* could be calculated using:(2)$$\bar{y_{a}}=\overset{10}{\underset{i=1}{\sum }}y_{a,i}/10$$

Figure 12a illustrates the relationship between *t_a_* and *d_m_*, which exhibited an obviously linear relationship. If a straight-line fitting was applied, the fitting line was *y* = 2.772*x* + 77.84, with a coefficient of determination of 0.9998. This allowed us to easily evaluate the distance using the processed data. Figure 12b shows the relationship between *y_a_* and *d_m_*. The inversely proportional relation gave another criterion to judge distance. The resulting fitting curve was $y=8.875\times 10^{-3}x^{-1.0475}$ with a coefficient of determination of 0.9973.

We characterized the maximum service range of the designed PMUT up to 35 mm. That is, using the proposed PMUT–microphone–DAQ system, the testing range within 35 mm could be identified using the calibration curve shown in Figure 11. Finally, two arbitrary distances of 17.75 and 22.26 mm were set. The two distance points were substituted into Equations (1) and (2) to derive time values of 126.239 and 139.072 μs, and strengths of 449.6 and 344.4 μV, respectively. The results were compared with the two distance test points by directly substituting them into the fitting equations. The data showed that the respective errors were 0.63% and 0.34% for the time-of-flight method, and 3.10% and 0.10% for the strength variation method.

## 6. Conclusions

Unlike the conventional PMUT fabrication process, a bottom-up fabrication technique, which started with growing a piezoelectric thin film on a patterned metal foil, selectively depositing an electrode layer to define the active region, and anchoring on a designed base, was proposed and successfully implemented. The fabricated PMUT possessed multiple resonances for operation and the fourth resonance was utilized as an ultrasonic emitter. Paired with the MEMS microphone module, a signal-processing scheme was developed to extract the pattern from the received signal for range-finding applications. Employing this pattern, along with time-of-flight and strength variation techniques to calculate the distance, an error of less than 0.7% was achieved for the time-of-flight method. It may be interesting to investigate that the properties of waves generated by the PMUT propagated into the underneath substrate in the future. The phase velocity, directionality, and acoustic intensity of the waves operated at different modes can be further studied.

## Figures and Tables

**Figure 1 sensors-19-04696-f001:**
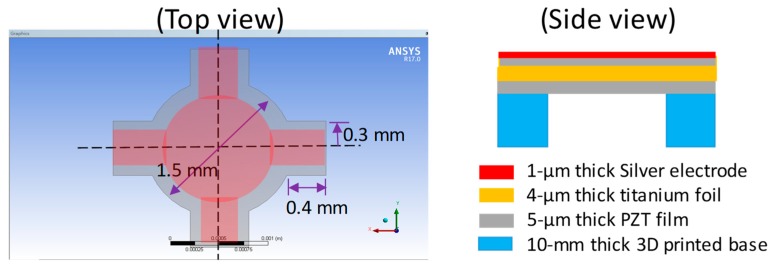
The designed structure of the PMUT vibrational portion (**left**) and the related structural parameters (**right**).

**Figure 2 sensors-19-04696-f002:**
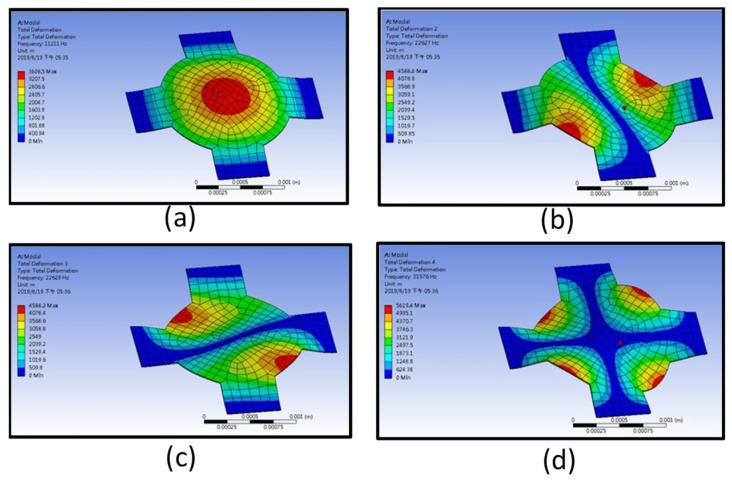
Finite element analysis (FEA) results: (**a**) the fundamental mode shape, (**b**) the second order harmonic mode shape, (**c**) the third order harmonic mode shape, and (**d**) the fourth order harmonic mode shape.

**Figure 3 sensors-19-04696-f003:**
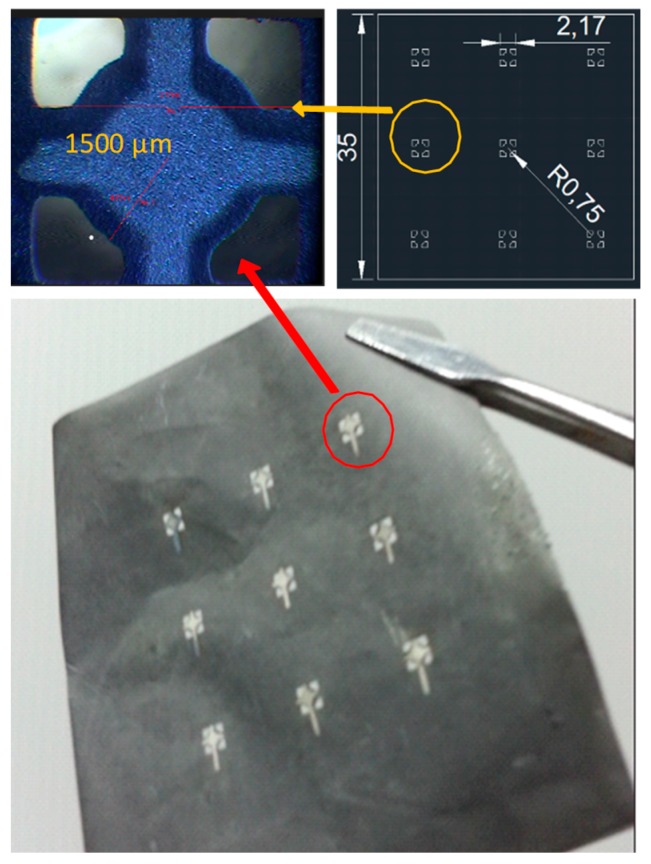
Results of the fabricated PMUT film element array on the patterned titanium foil.

**Figure 4 sensors-19-04696-f004:**
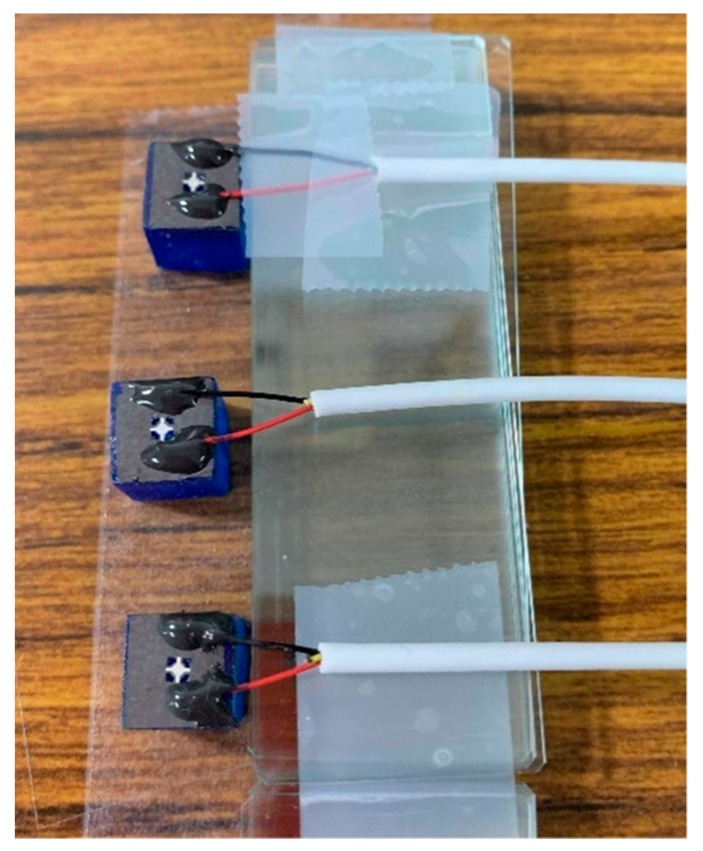
Complete PMUT device with the piezoelectric active film secured on the designed 3D printed base.

**Figure 5 sensors-19-04696-f005:**
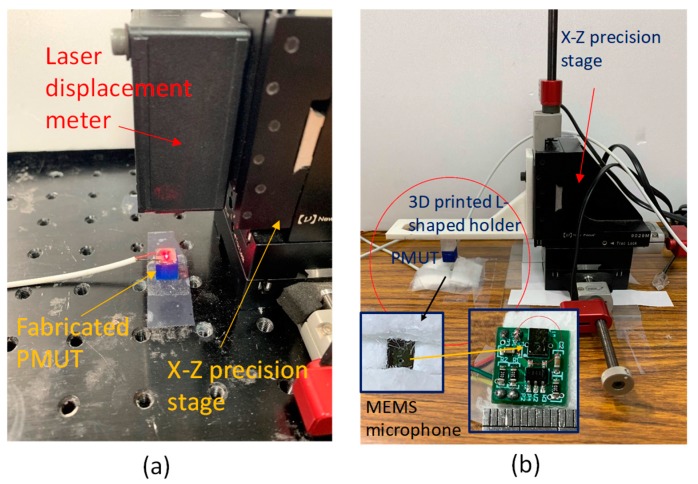
(**a**) Experimental setup for characterizing the PMUT performance. (**b**) Experimental setup for characterizing the PMUT–microphone pair for a range-finding application.

**Figure 6 sensors-19-04696-f006:**
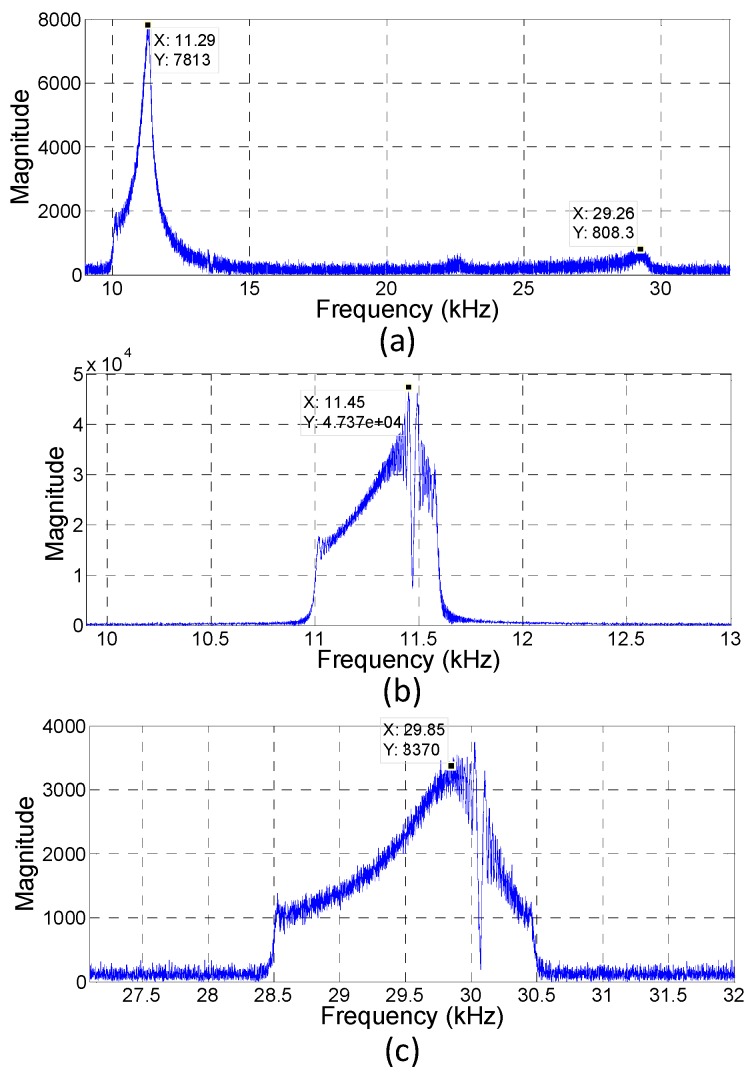
(**a**) The frequency response of the fabricated PMUT within 10−30 kHz to find the resonances. (**b**,**c**) Smaller sweeping ranges were used to locate the more accurate fundamental resonant frequency and fourth order resonance of the PMUT, respectively.

**Figure 7 sensors-19-04696-f007:**
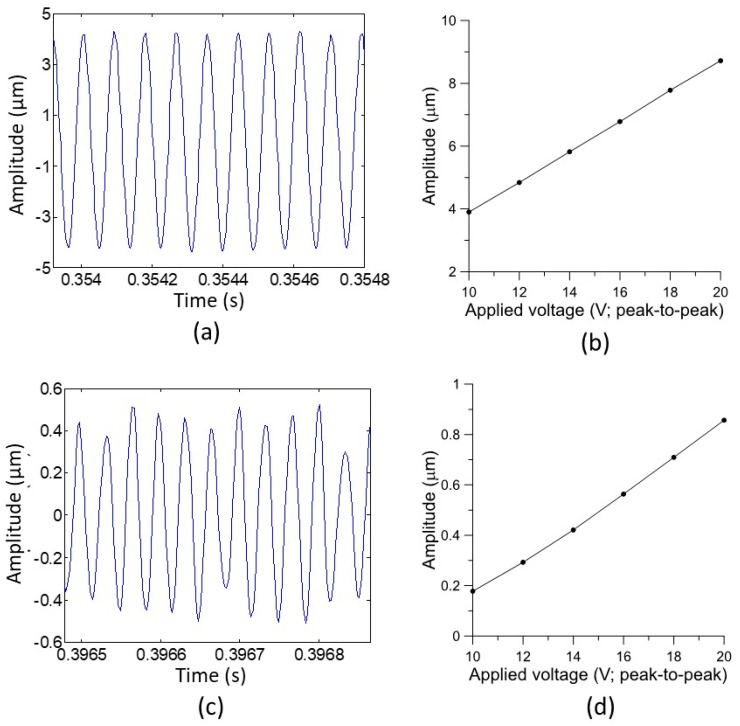
(**a**,**c**) Results of the measured displacement of the PMUT as a function of time with a 20 Vpp sinusoidal input voltage while being operated at the fundamental resonance and the fourth order resonance, respectively. (**b**,**d**) Results of measured displacements as a function of input voltages while being operated at the fundamental resonance and fourth order resonance, respectively.

**Figure 8 sensors-19-04696-f008:**
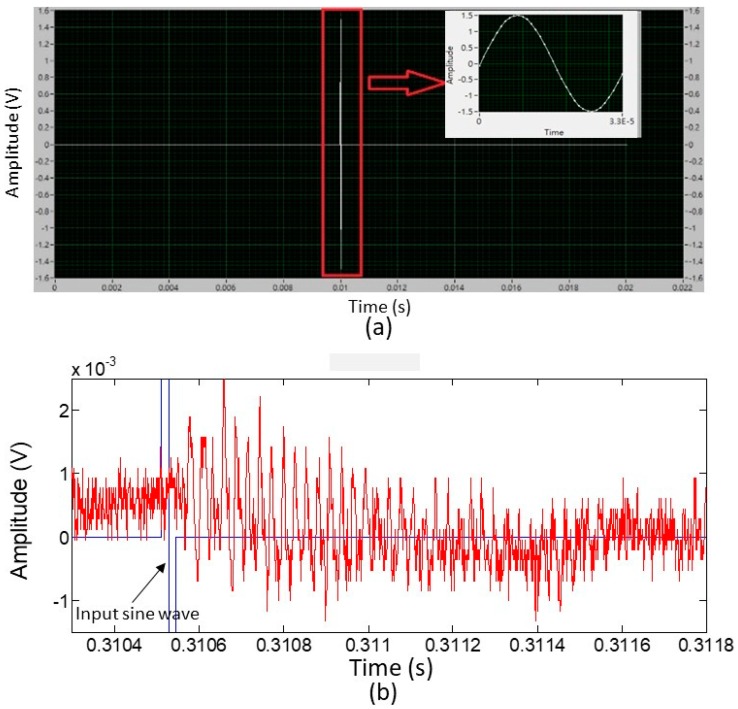
(**a**) The designed single waveform of the input signal to actuate the PMUT (image from the Labview software). (**b**) Acquired raw signal data from the microphone module at a distance of 10 mm away from the PMUT.

**Figure 9 sensors-19-04696-f009:**
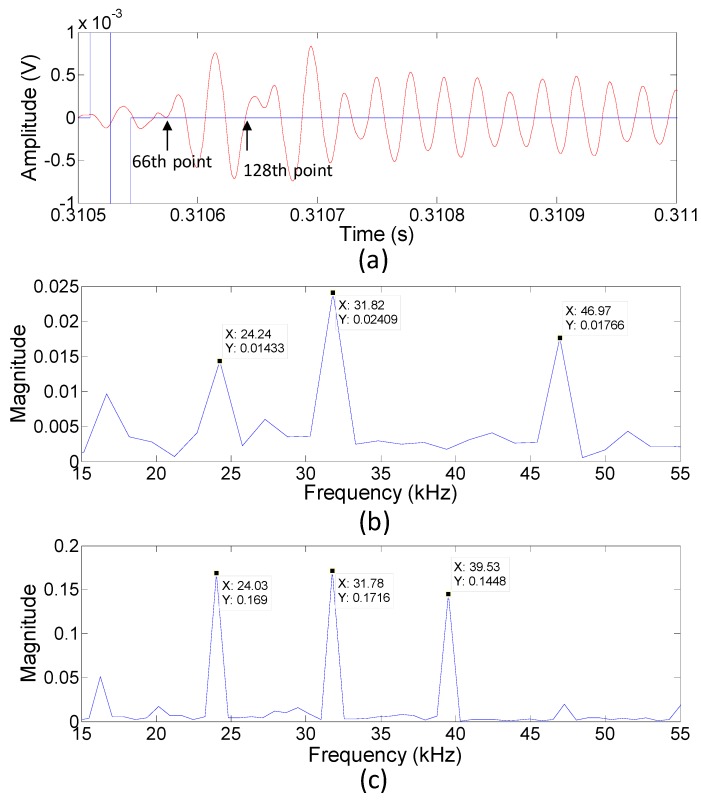
(**a**) The raw signal from the microphone processed with a third-order Butterworth bandpass filter. (**b**,**c**) The frequency responses of the processed microphone signal from the occasion at the beginning of the input signal to the positions of the 66th point and 128th point, respectively.

**Figure 10 sensors-19-04696-f010:**
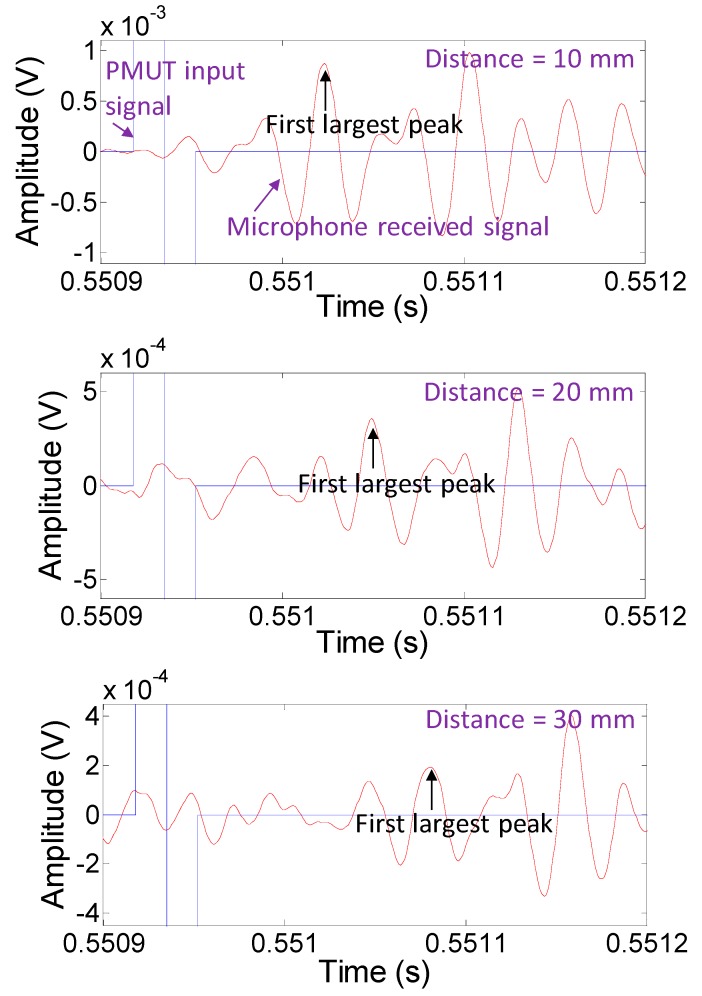
Results of the measured data for different distances after the bandpass filter processing. The first largest peaks as marked in the subplots displays were gradually delayed as the distance increased.

**Figure 11 sensors-19-04696-f011:**
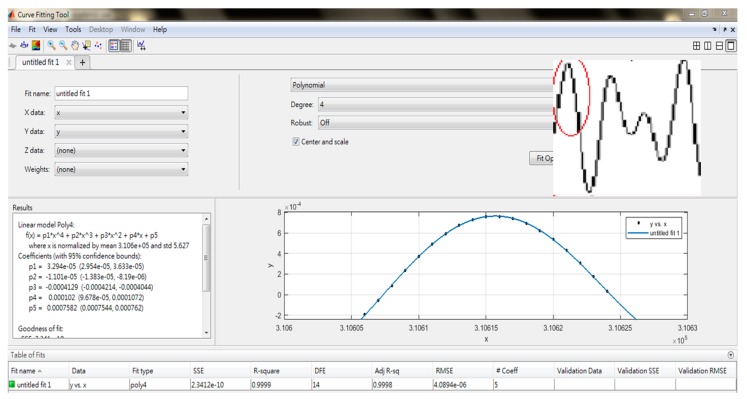
Utilizing a curve fitting method to find the peak time from the processed signal to increase the time resolution for range finding.

**Figure 12 sensors-19-04696-f012:**
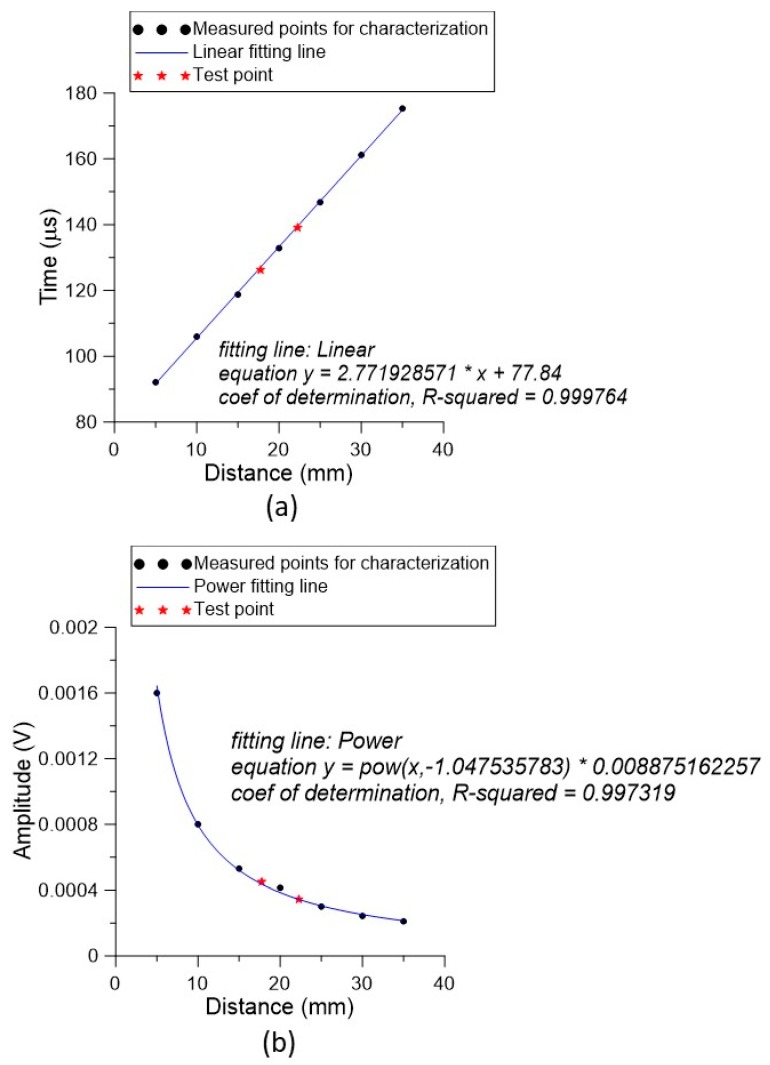
Characterized and tested results of the developed PMUT-microphone pair for range finding: (**a**) based on the time-of-flight principle and (**b**) based on the strength variation technique.

**Table 1 sensors-19-04696-t001:** Parameters used in the ANSYS simulation.

Layers	Young’s Modulus (GPa)	Density (kg/m^3^)	Thickness (µm)
Silver	60	8000	1
PZT	35	7500	4
Ti	110	4506	5
PZT	35	7500	4
Equivalent	43.45	6466	14

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
