# Peer review of "Piezoelectric Micromachined Ultrasonic Transducers with a Cost-Effective Bottom-Up Fabrication Scheme for Millimeter-Scale Range Finding"

_sensors, 2019, doi:10.3390/s19214696_

Round 1

Reviewer 1 Report

This paper presents a piezoelectric micromachined ultrasonic transducer (PMUT) fabricated on a metal foil, in which a bottom-up micromachining technique is used. A series of analyses and experiments are performed in sequence. The claimed application is also clear. However, it is not suitable for publication in its currents form. It is strongly suggested to reconsider the following comments.

In the design of PMUT, the final structural parameters are seem to be selected randomly. How to determine the structural parameters? I.e. what’s the design consideration? Even though the structure of the PMUT and corresponding base are described in Line 84 to 87, 94 to 95 and so on, a diagram of them is strongly suggested to show them in a clear way. Moreover, all components of the PMUT should be labeled and their key structural parameters should be pointed out. 1(a) shows the vibration portion of the PMUT, it is suggested that to combine this with the diagram recommended in above comment. The rest Fig. 1(b), (c), (d) and (e) are suggested to set as a figure for exhibiting FEM analyses results in the next. In line 122, it was claimed that the forth mode shape was supposed to be relatively stable during the operation since it occurred at a single frequency? In fact, the first mode shape also occurred at single frequency. Why the author chose the forth as the operation frequency? Some figures or plots should be revised to a clear way. For example, some arrows can be added in Fig. 4 to clear its specific indication. The current Fig. 5, Fig. 6, Fig. 7, Fig. 8, Fig. 9 are not clear and non-professional (format of size, font; some ticks in horizontal axis are seem to be added at will). The professional software, ORIGIN, MATLAB and so on, are strongly suggested to make data plots in more formal way instead of directly showing printscreen. In the fabrication of the PMUT, the chemical process is applied. It is the claimed bottom-up micromachining technique in this paper. If so, how to guarantee the designed structural dimensions within an acceptable range by using this process method? Moreover, what is fabricated error level of the structural dimensions? It can be seen that the leads are fixed on the prototype of the PMUT with some kind of adhesive. If so, what’s the used adhesive and whether it is conductive? In line 218, the author say the upper limit of the receiving frequency bandwidth of the selected MEMS microphone is 30 kHz. However, the following experiments show the PMUT operates at 28.95 kHz would be a better choice (line 291). The availability of the selected microphone should be commented because the claimed better operation frequency is so close to the upper receiving frequency limit. Line 278, 1.045 should be revised to 10.45 kHz. It is suggested that the horizontal axis in Fig. 7(b), Fig. 8(a) and Fig. 9 should be revised to time instead of data number. In Fig. 8(b) and (c), the curves are so unsmoothed maybe due to too few data points. This should be carefully considered and plotted with more data points.

Reviewer 2 Report

This paper reported an interesting study on the design, fabrication, and testing of PMUT exploiting bottom-up micromachine technique and 3-D printing. While the concept is interesting and understandable, the reported work lacks rationales on critical technical decisions and in-depth discussion of results. Thus, the manuscript is not recommended for publication, and please consider the following issues if the authors plan for resubmission or submission to other journals.

A. Technical issues

A1. Page 4, first paragraph, it seems that the authors use the equivalent parameters of the composite structure as the model parameters for the ANSYS model. It is not difficult to simulate the composite structure with multi-layer parameters, based on the existing model. How well does that agree with the equivalent model? Will the authors obtain a better match between the model and experimental results? Please comment on that.

A2. Page 5 line 171, the authors mentioned the sensor element was fabricated with near-zero residual stress. However, no evidence is provided to support this claim. If there is near-zero residual stress, to what extent such design would support improved performance? Also, would the following operation, such as electrodes and etching, introduce extra residual stress? Please comment on this.

A3. Page 6 line 201, the authors mentioned the input amplitude of 20 V peak-to-peak. Is this reasonable for IoT applications? In the cases of wearable devices, low power consumption and energy harvesting are required. On the other hand, microphone amplification of 1.5 times does not seem to be sufficient for meaning usage, considering the poor signal to noise ratio with only 10mm spacing in Fig. 7(b). Please provide evidence to support the choice of testing parameters.

A4. Fig. 5 and Fig. 8 need to be improved in terms of figure quality and font size. The markers within the plots are hardly visible.

A5. The end goal of this study is to develop an air-coupled ultrasound transducer. It is not clear the maximum service range the designed PMUT can support. Please specify that.

A6. Page 9, line 29. The rationale of choosing 29.85 kHz rather than 11 kHz is unclear. Considering the mode shape, it is expected to find the maximum film displacement at the first resonance. If the authors find it coincides with the audible sound range, the geometry shall be re-designed to move it to the ultrasound range, rather than stick with the weak mode and struggle with signal to noise ratio for later work. Please comment on this issue.

A7. Considering the poor SNR, the choice of bandpass filter for further interpretation is understandable. However, in page 11, line 363, why is the ‘receiving wave could be considered to be within the time interval between the 66th and 128th points’? What are typical methods to determine the time of flight for wave propagation? Is the currently adopted method commonly used? Why did the authors choose the current method rather than hundreds of other methods? Arbitrarily picking the starting point of ‘signal pattern’ as shown in Fig. 9 is simply unacceptable.

B. Technical writing and formatting

B1. This draft needs to be polished entirely both in technical writing and reasoning.

Reviewer 3 Report

The article “Piezoelectric Micromachined Ultrasonic Transducers with Cost-effective Bottom-up Fabrication Scheme for Millimeter-Scale Range Finding” describes a new piezoelectric micromachined ultrasonic transducer (PMUT) fabricated on a metal foil and associated with a MEM’s microphone for range-finding performance.

This paper is logically organized and clearly structured. The purpose and significance of the study are clearly stated and the research method is appropriate. Beside, theoretical and experimental results are presented explicitly by the figure and pictures and the author gives a comprehensive analysis in the discussion.

The identification of the parameters is done on the basis of experimental data. This is a real added value of the paper because the experimental/theoretical comparison is quite interesting. The experimental work is very well explained and brings a lot to the understanding of the research.

Despite some corrections have to be done, the paper presents an original work which can be published in Sensors Journal after minor revisions.

The corrections concern the presentation of the paper together with the description of the experimental and theoretical results. The list of improvements is as follow:

Introduction:

Line 27: Thousands should be used rather than hundreds (acoustic microscopy used more than 1Ghz waves) Line 44: SOI should be explained in full text line 64: dot after custom-made “characteristics”

Paragraph 2:

Line 68: FEA should be explained in full text. Line 122: A sentence is needed to explain why the fourth mode is so interesting, despite the small amplitude of the mode.

Paragraph 3:

Line 160: What about the Curie point at a 180°C temperature?

Paragraph 5:

Line 273: Is the displacement normal to the surface? It should be said. Line 274: pressure is not necessary Line 317: 10-6 should be 10-6

Conclusion:

In the conclusion a small discussion on the evolution of the work should be realized.  Because the authors as made a FEA simulation of the PMUT vibration modes, it would be interesting to simulate the conversion of the mechanical waves under the PMUT in order to study the phase velocity of the propagation of the waves inside the wafer. It would provide a better estimation of the velocity of the wave. But this work is considerable and will need another paper to be presented.

Question to the authors:

When you look at the figure 8C, it is obvious that the gap between the peaks is 7.75 KHz. The frequency space between the peaks is correlated to the time of flight of the emitted wave using the simple formula F= 1/T due probably to the multiple reflections. So 1/7.75 KHz gives you a time of flight of 129 µs which corresponds to a distance of 17.75 mm (see line 404). But figure 8 have been measured with a distance of 10 mm (see line 327). Could you confirm that figure 7 corresponds to a 10 mm spacing?

Round 2

Reviewer 2 Report

The authors respond to concerns appropriately.